



# Elements of future snowpack modeling - part 1: A physical instability arising from the non-linear coupling of transport and phase changes

Konstantin Schürholt[1], Julia Kowalski[2,3], and Henning Löwe[1]

[1]WSL Institute for Snow and Avalanche Research SLF, Flüelastr. 11, 7260 Davos, Switzerland
[2]AICES Graduate School, RWTH Aachen University, Schinkelstr. 2a, 52062 Aachen, Germany
[3]Computational Geoscience, University of Göttingen, Goldschmidtstr. 1, 37077 Göttingen, Germany

**Correspondence:** Henning Löwe (loewe@slf.ch)

**Abstract.** The incorporation of vapor transport has become a key demand for snowpack modeling where accompanied phase changes give rise to a new, non-linear coupling in the heat and mass equations. This coupling has an impact on choosing efficient numerical schemes for one-dimensional snowpack models which are naturally not designed to cope with mathematical particularities of arbitrary, non-linear PDE's. To explore this coupling we have implemented a stand-alone finite element solution of the coupled heat and mass equations in snow using FEniCS. We solely focus on the non-linear feedback of the ice phase exchanging mass with a diffusing vapor phase with concurrent heat transport in the absence of settling. We demonstrate that different, existing continuum-mechanical models derived through homogenization or mixture theory yield similar results for homogeneous snowpacks of constant density. For heterogeneous situations in which the snow density varies significantly with depth, we show that phase changes in the presence of temperature gradients give rise to a non-linear advection of the ice phase that amplifies existing density variations. Eventually, this advection triggers a wave instability in the continuity equations. This is traced back to the density dependence of the effective transport coefficients as revealed by a linear stability analysis of the non-linear PDE system. The instability is an inherent feature of existing continuum models and predicts, as a side product, the formation of a low density (mechanical) weak layer on the sublimating side of an ice crust. The wave instability constitutes a key challenge for a faithful treatment of solid-vapor mass conservation between layers, which is discussed in view of the underlying homogenization schemes and their numerical solutions.

## 1   Introduction

Neglecting vapor transport in the overall mass balance of a snowpack is considered as a serious uncertainty in snow modeling. As hypothesized in recent work on shallow tundra snowpacks (Barrere et al., 2017; Domine et al., 2016) persistent temperature gradients throughout the season may contribute to the depletion of snow density at the bottom of the snowpack due to persistent upward vapor fluxes. This problem is relevant for applications e.g. in permafrost where temperature gradients may induce the drying of soils with a feedback on snow metamorphism in the adjacent bottom layer (Domine et al., 2016), in turn affecting the ground thermal regime. Other applications of vapor transport comprise post-depositional re-distribution of stable water



isotopes in polar snow (Touzeau et al., 2018), density variations in polar firn (Li and Zwally, 2004), or the general impact on stratigraphy and metamorphism (Sturm and Benson, 1997).

The governing equations of macroscopic vapor transport in snow are used for a long time. The homogenized equations of heat and purely diffusive vapor transport including phase changes have been derived from mixture theory in early work (Adams and Brown, 1990; Morland et al., 1990; Bader and Weilenmann, 1992). More recently, the equations were re-derived from a rigorous two-scale expansion (Calonne et al., 2014) yielding the same form of the resulting equations. The asset of the latter approach is the parameter control, which allows to assess the model's scope and limits of applicability from the scale
analysis. This homogenization method was later generalized to include effects of thermal convection (Calonne et al., 2014). Lastly (Hansen and Foslien, 2015) was revisiting the problem of coupled heat and vapor transport using mixture theory which led to a more restrictive set of transport equations that rely on the assumption that the vapor concentration is always close, but not exactly in equilibrium with temperature. While the existing vapor schemes largely differ in the form of the effective transport coefficients, there is a general agreement on the basic type and form of the partial differential equations (PDE), that
govern coupled heat and diffusive vapor transport in snow. These PDEs are coupled, non-linear reaction diffusion equations. The diffusion terms are characterized by the effective diffusion constant and effective thermal conductivity in snow while the reaction (or source) terms describe the phase changes, i.e. the volume averaged, solid-vapor re-crystallization rates from metamorphism (Krol and Löwe, 2018). However, (Calonne et al., 2014) and (Hansen and Foslien, 2015) both neglect the feedback of phase changes through an *evolving* ice phase in their numerical experiments. It is this coupling that needs to be
understood for the incorporation of published homogenized vapor schemes into snowpack models for assessing the impact on snow density.

    The first attempt to solve the vapor diffusion equation in a snowpack model was recently undertaken by (Jafari et al., 2020) who equipped the model SNOWPACK with a vapor transport scheme as a non-linear reaction-diffusion equation. The numerical solution requires time-steps of 1 min and mesh sizes of 1 mm to avoid "numerical oscillations" that were observed,
even within an implicit, unconditionally stable numerical scheme. "Sawtooth effects" attributed to "slight numerical errors" were already revealed in early numerical work on the coupling of ice, vapor, and energy transport in snow (Adams and Brown, 1990) and likewise, the numerical solution in (Hansen and Foslien, 2015) shows oscillations if longer simulation runs are considered (personal communication with A. Hansen). Numerical issues have also been reported for solid-liquid phase changes when incorporating water flow into SNOWPACK via the Richardson equation which were attributed to density inhomogeneities
(Wever et al., 2014). Phase change processes in seasonal and polar snowpacks are commonly of interest on long time scales ideally using coarse meshes and large time steps to meet requirements for climate modeling. It is therefore necessary to understand the mathematical complexity of phase changes coupled to the mass transport equation. This will enable to either design stable and accurate numerical schemes or otherwise accept simplified treatments which, in the case of vapor transport, could be e.g. based on a strict equilibrium assumption (Li and Zwally, 2004; Touzeau et al., 2018).

Due to the lack of analytical solutions for these non-linear problems, confidence in orders of magnitudes of computed numbers can only be achieved via careful numerical experiments to address solver accuracy or mesh effects. This is naturally cumbersome within a full snowpack model. In addition, an explicit solution of the ice-mass conservation equation is com-





monly avoided by using the Lagrangian frame of reference of the settling equation (Brun et al., 1989; Lehning et al., 2002). This complicates the assessment of phase change models in their originally published Eulerian form. Stand-alone numerical in-

vestigations of specific model components are therefore justified and necessary to understand non-linear effects and numerical requirements for the desgin of efficient solvers in future snowpack models.

It is the aim of the present paper to advance the understanding of coupled heat and mass transport in snow by a careful numerical analysis of existing homogenization schemes. To overcome the limited flexibility in existing snowpack models we have implemented a stand-alone solver for the PDEs using the Finite Element (FE) framework FEniCS (Alnæs et al., 2015). The

python-based computing platform was previously used for other problems in cryospheric sciences (Cummings, 2016). We focus on the non-linear feedback of heat and vapor transport on an evolving ice phase. We consider the simplest setting and neglect convection in the gas phase and settling in the solid phase to provide a sound reference for future extensions. The question how these effects can be coupled to settling will be addressed in a companion paper (Simson et al., 2021). Our approach will reveal the rich mathematical complexity that is hidden in published models and will provide evidence that the origin of this

complexity is the density dependence of the effective transport coefficients in the presence of phase changes. We will show that this coupling causes the formation of wave patterns in the ice volume fraction profile as a true mathematical feature of the non-linear PDE system. This is confirmed by an analytical, linear stability analysis which relates unstable behavior to the density dependence of the effective (heat and mass) diffusion constants. The results suggest that previously observed oscillations in the numerical treatment (Adams and Brown, 1990; Jafari et al., 2020) were not numerical problems but may rather have signalled

emergent physics. With this work we seek to contribute to an understanding if and how these features should be taken into account in future work.

The paper is organized as follows. In Sec. 2 we state the governing partial differential equations from the homogenization schemes (Calonne et al., 2014) and (Hansen and Foslien, 2015). In Sec. 3 we outline the Finite Element solution of the weak formulation of the problem and its implementation in FEniCS. In Sec. 4 we first provide an inter-comparison of the two

models in their original test scenarios and the scenario of a thin "Gaussian crust" as a smooth density heterogeneity. In Sec. 5 we characterize the observed migration of the ice phase under vapor re-crystallization by comparing the full model with an approximate advection equation for the ice phase which is derived under simplified assumptions. In Sec. 6 we detail the wave patterns that emerge in the numerical solution of the Gaussian crust. The analysis of the mesh-resolution, integration time step and the residuals indicates that the wave patterns are intrinsic features of the mathematical model and not numerical artefacts.

This is confirmed by the linear stability analysis in Sec. 6.2. After a few sensitivity tests we will discuss practical consequences of our study for vapor transport modeling in snow models (Sec. 8) and provide summarizing conclusions.

## 2 Homogenized heat and mass transport

As a theoretical starting point we focus on two recently published homogenized formulations for an evolving vapor phase, namely (Calonne et al., 2014) and (Hansen and Foslien, 2015). Despite differences in their theoretical homogenization ap-





proach, asymptotic expansion (Calonne et al., 2014) and mixture theory (Hansen and Foslien, 2015), both models have a very similar mathematical structure that resembles earlier work of (Bader and Weilenmann, 1992).

## 2.1 Vapor scheme from (Calonne et al., 2014)

The two-scale expansion for (vapor) mass and energy (Calonne et al., 2014) leads to the following set of equations for the vapor density $\rho_v$ and the temperature $T$

$$(1-\phi_i)\frac{\partial}{\partial t}\rho_v - \nabla \cdot D_{\text{eff}}\nabla\rho_v \quad = \quad -\rho_i\, s\overline{v_n} \tag{1}$$

$$(\rho C)_{\text{eff}}\frac{\partial}{\partial t}T - \nabla \cdot k_{\text{eff}}\nabla T \quad = \quad L\, s\overline{v_n} \tag{2}$$

Here $\phi_i$ is the ice volume fraction, $D_{\text{eff}}$ the effective diffusion coefficient, $\rho_i$ is the density of ice density (assumed to be constant), $s$ the ice-air interface area per unit volume, $\overline{v_n}$ the volume averaged interface surface normal velocity indicating ice growth or decay, $(\rho C)_{\text{eff}}$ denotes the effective volumetric heat capacity (times snow density), $k_{\text{eff}}$ the effective thermal conductivity and $L$ the latent heat of sublimation.

The surface area density $s$, reflecting the current state of the microstructure and principally evolving in time, is assumed to be constant here in accordance with (Calonne et al., 2014).

The source term $-\rho_i\, s\overline{v_n}$ on the right hand side (r.h.s.) of the vapor equation (1) quantifies phase changes, i.e. the net condensation rate. This implies the form of the energy source term $L\, s\overline{v_n}$ through latent heat. In the simplest setting (Calonne et al., 2014), the volume averaged interface velocity is given by

$$\overline{v_n} = \frac{1}{\beta\rho_v^{\text{eq}}(T)}(\rho_v - \rho_v^{\text{eq}}(T)), \tag{3}$$

where $\beta$ is the inverse growth velocity and $\rho_v^{\text{eq}}(T)$ the saturation vapor density. Note that $\overline{v_n}$ strongly depends on the temperature, indicating the temperature feedback on the vapor mass balance. In order to close the system (1)-(3) parametrizations for the effective PDE coefficients must be provided (see below in section 2.3). The feedback of a spatio-temporally evolving ice phase $\phi_i(z,t)$ has not been considered in (Calonne et al., 2014).

## 2.2 Vapor scheme from (Hansen and Foslien, 2015)

A similar coupling scheme of the vapor transport to the energy equation has been put forward in (Hansen and Foslien, 2015). Using the same notation as above, these equations can be written in the form

$$(1-\phi_i)\frac{\partial\rho_v^{\text{eq}}}{\partial T}\frac{\partial T}{\partial t} - \nabla \cdot \left(D_{\text{eff}}\,\frac{\partial\rho_v^{\text{eq}}}{\partial T}\,\nabla T\right) \quad = \quad -c \tag{4}$$

$$(\rho C)_{\text{eff}}\frac{\partial}{\partial t}T - \nabla \cdot k_{\text{eff}}\,\nabla T \quad = \quad c\,\frac{L}{\rho_i} \tag{5}$$

Here the r.h.s. of both equations is expressed in terms of the condensation rate $c$.

The comparison of both models reveals that the vapor mass balance of (Hansen and Foslien, 2015) (4) can be obtained from the corresponding vapor mass balance of (Calonne et al., 2014) (1) using the following reasoning: The deviation of the vapor



concentration $\rho_v$ from its equilibrium value $\rho_v^{\mathrm{eq}}(T)$ will be mostly small. By assuming that the deviation $\rho_v - \rho_v^{\mathrm{eq}}(T)$ diffuses

faster than it relaxes locally, $\rho_v = \rho_v^{\mathrm{eq}}(T)$ is used in the diffusion term in (1), while a deviation from equilibrium is maintained

in the reaction term. Splitting the reaction term into an equilibrium contribution $\partial_t \rho_v^{\mathrm{eq}}$ and a perturbation, and using the known

temperature dependence of the saturation vapor density finally yields (4). (Hansen and Foslien, 2015) supports this derivation

with a (non-rigorous) scaling argument to avoid two conflicting equations for the temperature $T$.

Though both presented models are strikingly similar in structure, it should be emphasized that while system (1)-(3) is closed

and can be solved for $\rho_v$ and $T$, the same is not true for system (4) - (5) if we were to use the same condensation rate closure (3).

Due to the simplifying assumption outlined above, Hansen's model lacks an evolution equation for the (perturbed) vapor mass

balance. Since both, vapor mass balance (4), and energy balance (5) are actually formulated in terms of the temperature $T$,

(Hansen and Foslien, 2015) proceeds by consolidating both equations into a single one that no longer contains the condensation

rate

$$\left( (1 - \phi_i) \frac{\partial \rho_v^{\mathrm{eq}}}{\partial T} + \frac{\rho_i}{L} (\rho C)_{\mathrm{eff}} \right) \frac{\partial T}{\partial t} - \nabla \cdot \left( \left( D_{\mathrm{eff}} \frac{\partial \rho_v^{\mathrm{eq}}}{\partial T} + \frac{\rho_i}{L} k_{\mathrm{eff}} \right) \nabla T \right) = 0. \qquad (6)$$

The latter now poses a non-linear, yet closed PDE that can be solved for the evolving temperature. Back-substituting the

resulting temperature into either (4) or (5) allows to evaluate the condensation rate $c$, without remaining freedom to impose

yet another closure relation. An additional, important implication of the near-equilibrium assumption in (Hansen and Foslien,

2015) is the impossibility to impose arbitrary boundary conditions for the vapor phase. By construction of the model, the

boundary conditions are Dirichlet values for the equilibrium vapor concentration consistent with the boundary temperatures.

Similar to (Calonne et al., 2014) the feedback of a spatio-temporally evolving ice phase $\phi_i(z, t)$ was not been considered in

(Hansen and Foslien, 2015).

### 2.3   Parametrization of the PDE coefficients

Despite similarities in the forms of the PDEs, both models have used different parametrizations for the transport coefficients

$D_{\mathrm{eff}}$, $k_{\mathrm{eff}}$ and $(\rho C)_{\mathrm{eff}}$ and $\rho_v^{\mathrm{eq}}$. While the two-scale homogenization presented in (Calonne et al., 2014) contains *derived*

expressions for the effective properties in terms of the microstructure as a byproduct, mixture theory in (Hansen and Foslien,

2015) relies on independently *postulated* parametrizations. The implications of these differences are presently still under debate

(Fourteau et al., 2020).

In general, there is a broad agreement that all effective parameters are primarily influenced by the density or ice volume

fraction $\phi_i$. The goal of the present paper is not an exhaustive inter-comparison of different formulations of the parametrized

coefficients in the analysed snow pack models. We rather want to focus on the intrinsic features and physical implications of

the underlying process models. We therefore aim to keep differences due to specific flavours of the coefficients at the minimal

level. To this end we use the same parametrization for the equilibrium vapor pressure and the effective heat capacity in both

models but use different parametrizations for the effective thermal conductivity/diffusion constant. In the following we refer

to these PDE parametrizations as Calonne parametrization and Hansen parametrization. All coefficients are stated explicitly in

appendix A.





## 2.4 Feedback from an evolving ice phase

The consistent treatment of phase changes requires a dynamic ice phase that evolves through recrystallization alongside with the vapor phase in a mass-conserving way. This was neither considered in (Calonne et al., 2014) nor in (Hansen and Foslien, 2015). To investigate the feedback of an evolving ice phase on the two models from above we supply (1)-(3) and (4)-(5) with a dynamic ice mass conservation equation (Bader and Weilenmann, 1992; Krol and Löwe, 2018). In the absence of settling, but presence of phase changes the continuity equation reduces to an ordinary differential equation for each location in space

$$\frac{\partial}{\partial t}\phi_i = \begin{cases} s\,\overline{v_n} & \text{(Calonne)} \\ \frac{c}{\rho_i} & \text{(Hansen)}. \end{cases} \tag{7}$$

to balance the source terms in the vapor equations above. In Calonne's model $\overline{v_n}$ can be evaluated from the closure relation (3), whereas Hansen's $c$ has to be reconstructed from either (4) or (5) as outlined before.

In summary, all symbols and parameter values used in this study are provided in Table 1.

## 3 Finite Element solution in FEniCS

To minimize the coding overhead and focus on the physical problem while keeping access to advanced numerical adjustments, the coupled PDE model was implemented using the python-based Finite Element framework FEniCS (Alnæs et al., 2015). It provides a high-level API for the solution of PDEs in their weak formulation, with capabilities for parallelization, and cross-platform portability via docker images. FEniCS' flexibility arises from its interface to the Unified Form Language (UFL) (Alnæs et al., 2015) which supports an intuitive declaration of discrete FE formulations of variational problems.

Below we outline the spatial and temporal discretization for the weak formulation of both systems (1)-(3) and (4)-(5), and describe the solution strategy employed for the coupled PDEs.

## 3.1 Spatial discretization - Standard Galerkin

For the spatial discretization we note that the non-linear PDE systems, of interest can be formally rewritten in the form

$$F(u) = 0,$$

where $F(u)$ is a nonlinear differential operator and $u$ is the (vector valued) solution that comprises the unknown fields. Following a standard Galerkin approach, the PDE's weak formulation is obtained by multiplying $F(u) = 0$ with test functions $v$ and integrating over the domain. The Galerkin method then approximates the solution as the sum over a discrete set of basis functions $u_i$, viz $u \approx u_h = \sum_i u_i v_i$. This procedure allows to apply the differential operator to the test functions. The test functions have a small support only, which simplifies the integral evaluations significantly. By choosing different test functions $v = v_i$, the PDE system is reduced to finding the roots of a (non-linear) algebraic system. This standard Galerkin procedure is e.g detailed in (Donea and Huerta, 2003). The implementation into the open source finite element software FEniCS is straightforward.





**Table 1.** Symbols, defining equations and constants used in this study

| Symbol | Name | Equation/Value | Unit |
|--------|------|----------------|------|
| **PDE state variables** | | | |
| $T$ | Temperature | Eq. (2, 5) | K |
| $\rho_v$ | Vapor density | Eq. (1, 4) | $\text{kg m}^{-3}$ |
| $\phi_\text{i}$ | Ice volume fraction | Eq. (7) | – |
| **PDE coefficients** | | | |
| $(\rho C)_\text{eff}$ | Effective heat capacity | Eq. (A2) | $\text{J m}^{-3}\,\text{K}^{-1}$ |
| $D_\text{eff}$ | Effective vapor diffusion coefficient | Eq. (A3, A5) | $\text{m}^2\,\text{s}^{-1}$ |
| $k_\text{eff}$ | Effective thermal conductivity | Eq. (A3, A5) | $\text{W m}^{-1}\,\text{K}^{-1}$ |
| $c$ | Ice deposition rate | Eq. (5) | $\text{kg m}^{-3}\,\text{s}^{-1}$ |
| $\overline{v_n}$ | Averaged interface velocity | Eq. (3) | $\text{m s}^{-1}$ |
| $\rho_v^\text{eq}$ | Equilibrium vapor pressure | Eq. (A1) | $\text{kg m}^3$ |
| **Constant parameters** | | | |
| $\rho_i$ | Ice density | 917 | $\text{kg m}^{-3}$ |
| $k_i$ | Ice thermal conductivity | 2.3 | $\text{W m}^{-1}\,\text{K}^{-1}$ |
| $k_a$ | Air thermal conductivity | 0.024 | $\text{W m}^{-1}\,\text{K}^{-1}$ |
| $C_i$ | Ice heat capacity | 2000 | $\text{J kg}^{-1}\,\text{K}^{-1}$ |
| $C_a$ | Air heat capacity | 1005 | $\text{J kg}^{-1}\,\text{K}^{-1}$ |
| $L$ | Ice latent heat of sublimation | 2835332.6 | $\text{J kg}^{-1}$ |
| $D_0$ | Diffusion coefficient of vapor in air | 2e-5 | $\text{m s}^{-2}$ |
| **Parameters of the linear stability analysis** | | | |
| $T_\text{ref}$ | Reference temperature | Eq. (28)/263 | K |
| $\phi_\text{i,0}$ | Reference ice volume fraction | Eq. (29)/0.3 | – |
| $\alpha$ | Linearized kinetic coefficient | Eq. (26)/3.62 | $\text{m}^3\text{s}^{-1}\text{kg}^{-1}$ |

## 3.2 Temporal discretization - Theta method

Time derivatives are approximated using the theta method (also known as Rothe's method) (Donea and Huerta, 2003). For a PDE of the form

$$\frac{\partial}{\partial t}u = f(u) \tag{8}$$

the discretization reads

$$\frac{u^{n+1} - u^n}{\Delta t} = \theta f(u)^{n+1} + (1-\theta)f(u)^n + \mathcal{O}\big((1/2 - \theta)\,\Delta t, \Delta t^2\big) \tag{9}$$

where $u^n$ denotes the state vector's solution at time $t_n$. Note that for $\theta = 0$ and $\theta = 1$ this reduces to standard first order explicit and implicit Euler methods, respectively, whereas for $\theta \in (0,1)$ is constitutes a weighted average of both.





### 3.3 Coupling scheme

Due to the different time scales of the involved equations, a monolithically coupled solution for the vector $u = (T, \rho_v, \phi_i)$ would
be most consistent, yet turns out to be inefficient. The vapor equation has by far the fastest dynamics, followed by the energy.
The ice mass balance instead has a much slower dynamics. Since both water vapor and energy equation share the structure of
a diffusion-reaction equation and directly determine the r.h.s. of all equations, we solve them together, and split the solution to
the ice equation with its expected slower dynamics in a separate step. This operator splitting allows an independent fine-tuning
of the numerical methods. Practically, we included the r.h.s. of the equations as an algebraic constraint in the state vector,
i.e. $u = (T, \rho_v, \overline{v_n})$ for Calonne and $u = (T, c)$ for Hansen, which are then solved in a fully coupled way. The $\phi_i$-dependent
coefficients of the vapor and energy equation use $\phi_i$ from the previous time step. Subsequently $\phi_i$ is updated via $\overline{v_n}/c$. Time
step size is mostly determined through the vapor and energy dynamics. Changes in $\phi_i$, and hence in the PDE coefficients are
therefore expected to be very small, which justifies the splitting based coupling procedure.

We apply different theta values for each differential operator. For the diffusion operators in the vapor and energy equations
we use $\theta = 0.5$ (Crank-Nicolson) which is known to be stable and converge of second order for linear operators. As the phase
change induced source terms are expected to be stiff, a fully implicit, unconditionally stable Backwards Euler scheme with
$\theta = 1.0$ is applied.

## 4 Model comparison

To evaluate the models, we tested them on three different *scenarios* comprising specific combinations of initial conditions
(IC) and boundary conditions (BC). The first two scenarios discussed in sections 4.1 and 4.2 are taken from (Calonne et al.,
2014) and (Hansen and Foslien, 2015), respectively, while the last (section 4.3) is the simulation of a Gaussian shaped crust.
The three scenarios test different relevant aspects of snow modeling, namely the transient response to time dependent BCs,
piecewise-linear ice profiles covering the entire range of snow densities and lastly a high density layer in a small sample
with smooth, high density gradients. For each of the three physical scenarios, always evaluate three model formulations: First,
Calonne's equations (1), (2), (7) with Calonne coefficient closure from Appendix A (referred to Cal-Eq/Cal-Par). Second,
Hansen's equations (6), (7) with the Hansen coefficient closure from Appendix A (Han-Eq/Han-Par). Third, the mixed case of
Hansen's equations (A5), (7) with Calonne coefficient closure (Han-Eq/Cal-Par).

### 4.1 Scenario 1: Homogeneous snow - transient heating at the boundary

The first scenario is taken from (Calonne et al., 2014) who investigated the response of a homogeneous snow layer to transient
heating. To this end we use the IC

$$T(z, 0) = T_0 \tag{10}$$

$$\rho_v(z, 0) = \rho_v^{\mathrm{eq}}(T(z, 0)) \tag{11}$$

$$\phi_i(z, 0) = \phi_{i,0} \tag{12}$$



with $T_0 = 273K$ and $\phi_{i,0} = 0.3$. We employ a a fixed Dirichlet BC at $z = 0$ and a transient temperature drop at $z = H$, viz

$$T(0,t) = T_0$$

$$T(H,t) = T_0 - (T_H - T_0)\left(\frac{t}{\tau}\Theta(t) - \frac{t-\tau}{\tau}\Theta(t-\tau)\right) \tag{13}$$

$$\rho_v(0,t) = \rho_v^{\text{eq}}(T(0,t))$$

$$\rho_v(H,t) = \rho_v^{\text{eq}}(T(H,t))$$

where the transient drop is characterized by the Heaviside step function $\Theta$ with parameters $\tau = 5h$ and $T_H = 263K$.

For this combination of IC and BC we obtain the results in Figure (1) where the solutions of all three cases at $t = 10h$ are shown. Notably, Calonne and Hansen's models along with their own (different) parameterizations for the PDE coefficients deviate in all variables, since Hansen's effective diffusion constant is higher. This difference does not depend on the underlying process models, but is rather due to the used formulations of the PDE coefficients as shown by the mixed case (Han-Eq/Cal-Par) which coincides with Han-Eq/Han-Par. Only a small difference can be observed close to the right boundary, where the changes in $T$ and $\rho_v$ are the fastest and therefore the assumption $\rho_v = \rho_v^{\text{eq}}(T)$ underlying (Hansen and Foslien, 2015) is violated.

## 4.2 Scenario 2: Heterogeneous snow - fixed temperature boundary conditions

Next we investigate the test case envisaged by (Hansen and Foslien, 2015) that comprises a piece-wise linear density profile as an approximation for a layered snowpack. The IC is given by

$$
\begin{align}
T(z,0) &= T_0 - (T_H - T_0)\frac{z}{H}, \tag{14}\\
\rho_v(z,0) &= \rho_v^{\text{eq}}(T(z,0)) \tag{15}
\end{align}
$$

$$
\phi_i(z,0) = \begin{cases}
1 - 9.2425z & z \in [0, 0.08], \\
0.2606 & z \in [0.08, 0.64], \\
0.2606 + 4.915(z - 0.64) & z \in [0.64, 0.72], \\
0.6538 & z \in [0.72, 0.75], \\
0.6538 - 4.915(z - 0.75335) & z \in [0.75, 0.86], \\
0.1295895 & z \in [0.86, 1.0],
\end{cases} \tag{16}
$$

with $T_0 = 273K$ and $T_H = 253K$. The BC are given by

$$
\begin{align}
T(0,t) &= T_0, \\
T(H,t) &= T_H \tag{17}\\
\rho_v(0,t) &= \rho_v^{\text{eq}}(T(0,t)), \\
\rho_v(H,t) &= \rho_v^{\text{eq}}(T(H,t)).
\end{align}
$$

The corresponding simulation results are shown in Figure (2) at simulation time of 10 days. Again, the fields $\rho_v$ and $T$ highly agree in homogeneous regions as long as the same *paramerizations* for the PDE coefficients are used. In contrast, the phase





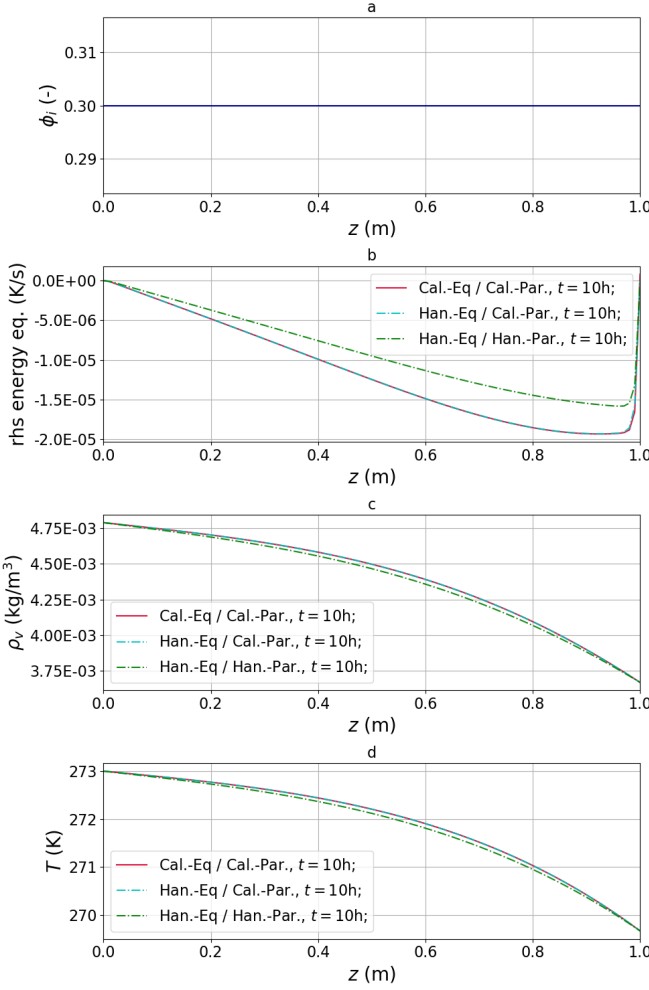

**Figure 1.** Comparison of simulation results conducted based on Calonne's and Hansen's model each with their own formulation of coefficients, as well as Hansen's model with Calonne's coefficient for the response to a transient temperature increase at the boundary.

change term (r.h.s. of the equations) shows an agreement only if the the same *equations* are used. This is revealed in the kinkregions of the profile where (Han-Eq/Han-Par) coincides with (Han-Eq/Cal-Par). This shows the different sensitivities of the

$T, \rho_v$ and the phase change terms on the form of the homogenized equation.

### 4.3 Scenario 3: The Gaussian crust

To further detail the response to high-gradient regions in the ice phase our third scenario investigates the response of the models to a smooth, small-scale density variation under a temperature gradient in a shallow snow sample of height $H = 0.02$m, as



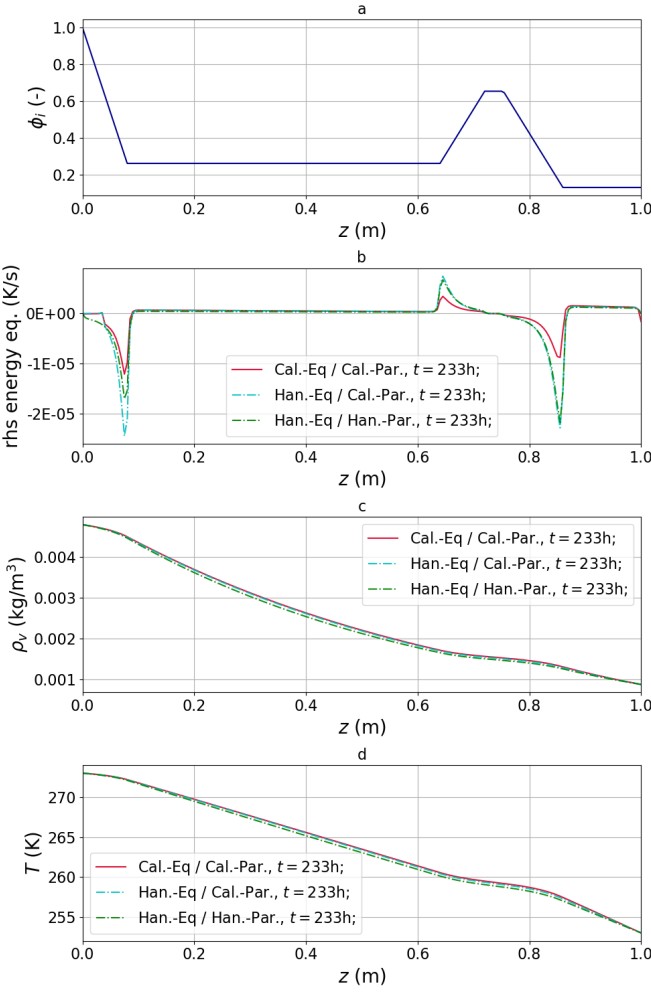

**Figure 2.** Comparison of simulation results conducted based on Calonne's and Hansen's model each with their own formulation of coefficients, as well as Hansen's model with Calonne's coefficient for the response to a piecewise-linear, layered profile.

typically encountered in tomography experiments (Hammonds et al., 2015). In the following we solely focus on differences

between (Cal-Eq/Cal-Par) and (Han-Eq/Han-Par) and investigate this difference at different physical times.

We refer to this scenario as a Gaussian crust. As IC we employ

$$T(z,0) \quad = \quad T_0 - (T_H - T_0)\frac{z}{H}, \tag{18}$$

$$\rho_v(z,0) \quad = \quad \rho_v^{\mathrm{eq}}(T(z,0)) \tag{19}$$

$$\phi_i(z,0) \quad = \quad \phi_{i,0} + \Delta\phi_{i,0} \exp -\frac{(z-z_0)^2}{2\sigma^2} \tag{20}$$

with $T_0 = 273\mathrm{K}$, $T_H = 253\mathrm{K}$, $\phi_{i,0} = 0.3$, $\Delta\phi_{i,0} = 0.2$, $z_0 = 0.01\mathrm{m}$, $\sigma^2 = 5 \cdot 10^{-7}\mathrm{m}$. For the Gaussian crust we use the same boundary conditions as in (17).





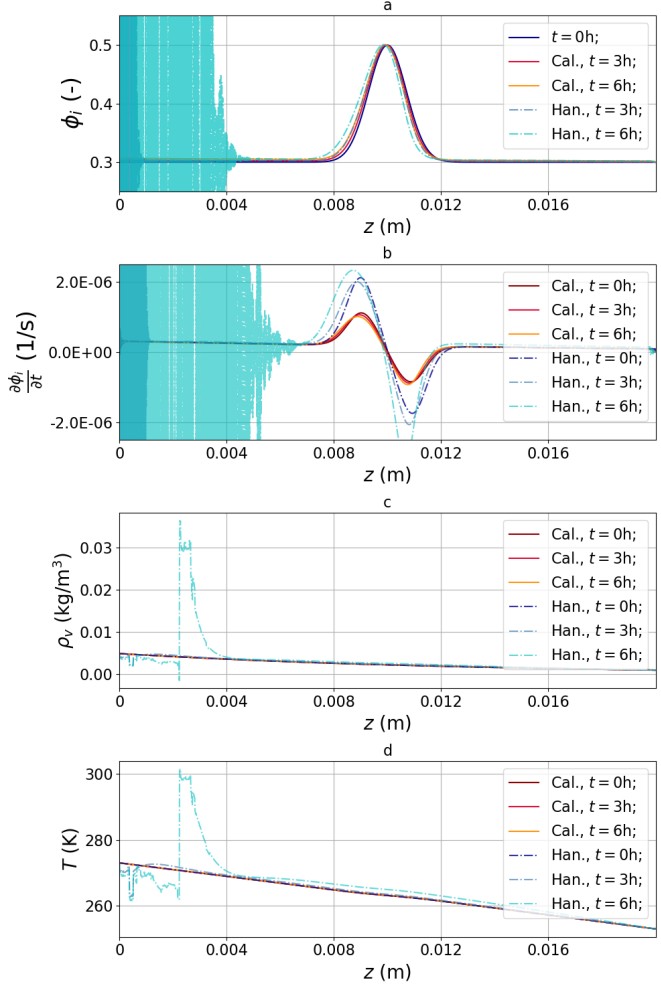

**Figure 3.** Comparison of simulation results conducted based on Calonne's and Hansen's model each with their own formulation of coefficients for the evolution of a Gaussian crust.

Figure (3) shows the simulation results for several points in time. Similar as before the solutions for $\rho_v$ and $T$ agree very well and notable differences are observed in density transition regions. From this comparison for a smooth density variation, two additional observations can be made, consistently for both models.

First, the Gaussian crust shows a quasi-advection towards the warm boundary, despite the absence of an explicit advection term in the ice equation. During this quasi-advection, density gradients steepen on the cold side and flatten on the warm side. Far away from the crust a linear density gradient emerges in the domain as a consequence of the boundary conditions. The difference between the models lies in the apparent advection velocity. This is consistent with the observed differences in the phase changes since recrystallization rate differ approximately by a factor of two.





Second, in both models oscillations emerge at the lower boundary. They are modest in the Calonne model, and arise at considerable later time with smaller magnitude. For the Hansen model these are apparent immediately.

These two observations are further detailed below.

## 5   Quasi-advection of the ice phase

The quasi-advection of density heterogeneities in the ice phase despite the absence of an explicit advection term in (7) is a
consequence of phase changes (vapor sublimation and re-sublimation) as an intrinsic feature of the coupled system.

To understand the origin of this quasi-advection, we approach the coupled system (1), (2) and (7) with several analytical approximations. Justified by the separation of time-scales we first restrict ourselves to stationary heat transfer and neglect the phase changes in the energy equation following arguments given in (Calonne et al., 2014). This is a considerable simplification since the two-way coupling between heat and vapor is eliminated. Consequently, the heat equation $\nabla k_{\mathrm{eff}} \nabla T = 0$ can be solved
for the BC (17) independent of the vapor transport process even for an inhomogeneous ice profile, with its exact solution given by

$$T(z) = T_0 + (T_H - T_0) \frac{\int_0^z dz'\, [k_{\mathrm{eff}}(\phi_i(z'))]^{-1}}{\int_0^H dz'\, [k_{\mathrm{eff}}(\phi_i(z'))]^{-1}} \tag{21}$$

The local temperature (and its gradient) is thus a functional of the heterogeneous, non-constant ice volume fraction $\phi_i$ via the dependence of the effective conductivity $k_{\mathrm{eff}}$ on $\phi_i$. We express this in the form

$$T = \mathcal{F}_T[\phi_i] \tag{22}$$

which defines the functional $\mathcal{F}_T$ via (21). Motivated by the similarity between the solutions of (Calonne et al., 2014) and (Hansen and Foslien, 2015) found in the previous section, we adopt the simplification from (Hansen and Foslien, 2015) and account for deviations of the vapor concentration from equilibrium only in the phase change (source) term. Again, we use a stationary diffusion equation $\nabla D_{\mathrm{eff}} \nabla \rho_v^{\mathrm{eq}} = \rho_i s \overline{v_n}$ due to the separation of time-scales between vapor and ice. This allows to
rewrite the ice mass conservation in a conservative form as a single advection equation

$$\frac{\partial}{\partial t} \phi_i = s \overline{v_n} \quad \Leftrightarrow \quad \frac{\partial}{\partial t} \phi_i + \nabla G[\phi_i] = 0, \tag{23}$$

with a functional

$$G[\phi_i] = D_{\mathrm{eff}}[\phi_i] \frac{\partial \rho_v^{\mathrm{eq}}(T)}{\partial T}\bigg|_{T = \mathcal{F}_T[\phi_i]} \frac{(T_H - T_0)[k_{\mathrm{eff}}(\phi_i(z))]^{-1}}{\int_0^H dz'\, [k_{\mathrm{eff}}(\phi_i(z'))]^{-1}} \tag{24}$$

that expanded $\rho_v^{\mathrm{eq}}(T)$ by means of the chain rule and substituted in the previously derived temperature expression (22). The
boundary conditions

$$\frac{\partial}{\partial t} \phi_i(z, t) = 0 \quad \text{for } z = 0, z = H \tag{25}$$





follow from the BC $\rho = \rho_v^{\mathrm{eq}}$ of the vapor phase.

   We stress the significance of this result. First, by using the approximations for the heat and vapor transfer, the three coupled
heat and mass conservation equations can be equivalently cast into a single continuity equation for the ice phase. Second,
the form of this continuity equation is a non-linear and non-local advection equation, which explains the nature of this quasi-
advection as a variant of shock formation reminiscent of the non-linear Burgers equation (Du et al., 2012).

   To test the derived approximation we have compared the numerical solution of (23) to the full solution of the Calonne model

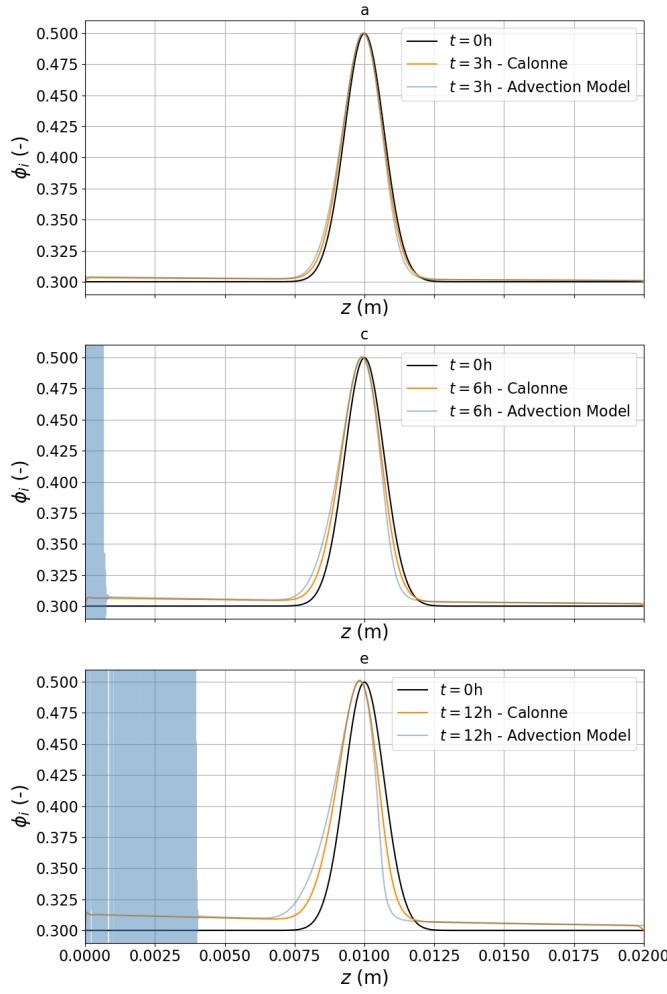

**Figure 4.** Comparison between the full homogenized model and the derived advection equation (23) for an initial Gaussian crust.

as shown in figure (4) for the Gaussian crust. In general, the agreement between the two models is very close, yielding almost
identical results in homogeneous regions, i.e. towards the boundaries and close to the center of the crust. In the high gradient
regions both models predict a steepening of the gradient, which is faster in the advection equation (21). Since (21) involves





the same quasi-equilibrium approximation as used for the Hansen model, these quantitative differences are consistent with the results from the previous section.

Despite the remaining differences, the approximation (21) allows to trace back unambigously the quasi-advection and gradient steepening to the dependence of the effective diffusion constant and the effective thermal conductivity on ice volume frac-
tion: If $D_{\text{eff}}$ was linear in $\phi_i$ and $k_{\text{eff}}$ was constant, (21) would take the form of a simple advection equation $\partial_t \phi_i + v \nabla \phi_i = 0$ with constant $v$. This would imply a shape invariant advection of any initial density profile. The fact that $D_{\text{eff}}$ decreases, and $k_{\text{eff}}$ increases with ice volume fraction, decreases the ice flux functional $G$ in high density regions over lower density regions. Both effects contribute in the same direction and explain the emergent asymmetry in the crust.

Similar to (3), the numerical solution of the advection equation also displays oscillations at the boundary, already after a few
hours. This will be further detailed in the following.

## 6  Pattern Formation

All numerical solutions so far are subject to oscillations "at some point in time" irrespective of the details. Therefore we will investigate the nature of these oscillations, now only considering the full Calonne model applied to the Gaussian crust.

### 6.1  Oscillatory solutions

The following example shows simulations with varying mesh size (number of elements $n_e$) and varying time steps $dt$ for the numerical solution at a fixed physical time. The results are shown in Figure (5). Figure (5)a shows the entire domain, revealing oscillations at the left boundary and on the sublimating side of the crust. Figure (5)b shows a close up of the left boundary while (5)c shows close-up of the sublimating side of the crust. For very low mesh resolution ($n_e = 100$, corresponding to 0.2mm) and large time step (1 min) the solution oscillates from node-to-node (a) while for sufficiently high resolution the
numerical solution converges to a smooth wave pattern independent of temporal and spatial resolution. This is interesting, as it suggests that these waves are true, intrinsic features of the full Calonne model equations, rather than an artefact of the numerical scheme. As an additional check we have analyzed the residuals of the numerical solution (cf. Appendix (B)) which confirms that patterns persist even when enforcing the residuals to approach zero.

### 6.2  Linear stability analysis

To comprehend the oscillatory nature of the solution we analyzed the problem theoretically within perturbation theory. Pattern formation in non-linear PDE systems can be generally understood by investigating the dynamics of perturbations around a known stationary state via linear stability analysis of the dynamical equations. To do so, we start from the full, non-linear,


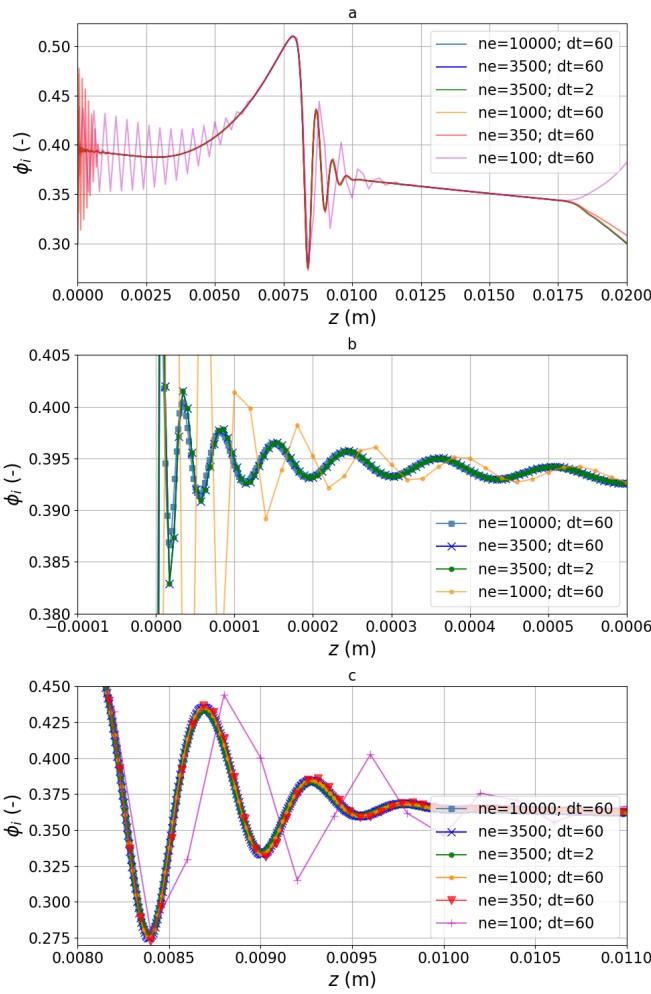

**Figure 5.** Sensitivity of the numerical solution to mesh resolution and time step size: (a) entire domain (b) zoom into the left boundary region (c) zoom into the sublimating (right) side of the Gaussian crust.

coupled, transient situation (Calonne model) in the form

$$
\begin{aligned}
(\rho C)_{\text{eff}} \partial_t T - \nabla_z k_{\text{eff}} \nabla_z T &= L\alpha(\rho_v - \rho_v^{\text{eq}}) \\
(1-\phi_i)\partial_t \rho_v - \nabla_z D_{\text{eff}} \nabla_z \rho_v &= -\alpha\rho_i(\rho_v - \rho_v^{\text{eq}}) \\
\partial_t \phi_i &= \alpha(\rho_v - \rho_v^{\text{eq}})
\end{aligned}
\tag{26}
$$


only subject to the simplification that on the r.h.s. the division by $\rho_{\text{eq}}(T)$ in Eq. (3) is subsumed in the prefactor $\alpha$ which is assumed to be constant.





The system (26) is, as before, equipped with Dirichlet BC for the two PDE's

$$T(0) = T_0$$
$$T(H) = T_H$$
$$\rho_v(0) = \rho_v^{\text{eq}}(T_0)$$
$$\rho_v(H) = \rho_v^{\text{eq}}(T_H) \tag{27}$$

at the bottom $z = 0$ and top $z = H$ of the domain. To proceed analytically, we need to make an additional assumption to

obtain an exact stationary solution of the system. To this end we will assume that the equilibrium vapor concentration is a linear function in $T$. This is reasonable within small layers where temperature differences are small and a linearization of the equilibrium vapor curve can be always justified. This linearization is written in the form

$$\rho_v^{\text{eq}}(T) = \rho_0^{\text{eq}} + \rho_1^{\text{eq}}(T - T_{\text{ref}}) \tag{28}$$

as an expansion around a reference temperature $T_{\text{ref}}$ (e.g. the mean temperature in the sample). With this assumption, a sta-

tionary solution of Eq.(26) can be obtained by inspection. It is easily verified that

$$T^{(0)}(z) := T_0 + (T_H - T_0)\frac{z}{H}$$
$$\rho_v^{(0)}(z) := \rho_v^{\text{eq}}(T^{(0)}(z))$$
$$\phi_i^{(0)}(z) := \phi_{i,0} \tag{29}$$

satisfies (26) with BC (27) for arbitrary, constant volume fraction $\phi_{i,0}$.

To carry out the stability analysis we use a vector notation and combine the fields in the vector $\boldsymbol{u} := (T, \rho_v, \phi_i)$. Then the PDE system (28) can be written in matrix form

$$\boldsymbol{C}(\boldsymbol{u})\partial_t\boldsymbol{u} - \boldsymbol{K}(\boldsymbol{u})\nabla_z^2\boldsymbol{u} - (\nabla_z\boldsymbol{K}(\boldsymbol{u}))(\nabla_z\boldsymbol{u}) = \boldsymbol{R}\boldsymbol{u} + \boldsymbol{f} \tag{30}$$





with state dependent $(3 \times 3)$ matrices $\boldsymbol{K}(\boldsymbol{u}), \boldsymbol{C}(\boldsymbol{u})$ constant matrix $\boldsymbol{R}$ and constant vector $\boldsymbol{f}$ defined by

$$\boldsymbol{C} = \begin{bmatrix} (\rho C)_{\text{eff}} & 0 & 0 \\ 0 & (1-\phi_i) & 0 \\ 0 & 0 & 1 \end{bmatrix} \tag{31}$$


$$\boldsymbol{K} = \begin{bmatrix} k_{\text{eff}} & 0 & 0 \\ 0 & D_{\text{eff}} & 0 \\ 0 & 0 & 0 \end{bmatrix} \tag{32}$$

$$\boldsymbol{R} = \begin{bmatrix} -L\alpha\rho_1^{\text{eq}} & L\alpha & 0 \\ \rho_i\alpha\rho_1^{\text{eq}} & -\rho_i\alpha & 0 \\ -\alpha\rho_1^{\text{eq}} & \alpha & 0 \end{bmatrix} \tag{33}$$

$$\boldsymbol{f} = \begin{bmatrix} L\alpha\rho_1^{\text{eq}}T_{\text{ref}} - L\alpha\rho_0^{\text{eq}} \\ -\rho_i\alpha\rho_1^{\text{eq}}T_{\text{ref}} + \rho_i\alpha\rho_0^{\text{eq}} \\ \alpha\rho_1^{\text{eq}}T_{\text{ref}} - \alpha\rho_0^{\text{eq}} \end{bmatrix} \tag{34}$$

$$\tag{35}$$

Then, the (stable) stationary state of (29) is denoted by the vector

$$\boldsymbol{u}^{(0)} := (T^{(0)}, \rho_v^{(0)}, \phi_i^{(0)}) \tag{36}$$

The non-linearities of (30) emerge from the dependence of the matrices $\boldsymbol{K} \equiv \boldsymbol{K}(\boldsymbol{u})$ and $\boldsymbol{C} \equiv \boldsymbol{C}(\boldsymbol{u})$ on $\boldsymbol{u}$. Note that this dependence is solely through the third component of $\boldsymbol{u}$ (i.e. $\phi_i$) which highlights the special role of the density (evolution).

Next we investigate the linear stability of the fixed point (36). To this end we make an ansatz

$$\boldsymbol{u} = \boldsymbol{u}^{(0)} + \boldsymbol{u}^{(1)} \tag{37}$$

and investigate the dynamics of the deviation from the stationary state $\boldsymbol{u}^{(0)}$ by deriving an equation for $\boldsymbol{u}^{(1)}$ through a perturbation expansion of (30) in first order. This procedure (carried out in detail in Appendix (C)) yields an equation for the Fourier modes $\tilde{\boldsymbol{u}}^{(1)}(k)$ of the perturbation for wave number $k$ which can be written in the form

$$\partial_t \tilde{\boldsymbol{u}}^{(1)}(k,t) = \boldsymbol{M}_k \tilde{\boldsymbol{u}}^{(1)}(k,t) \tag{38}$$

with a wave number dependent matrix

$$\boldsymbol{M}_k := -k^2 [\boldsymbol{C}^{(0)}]^{-1} \boldsymbol{K}^{(0)} + ik [\boldsymbol{C}^{(0)}]^{-1} \boldsymbol{V}^{(0)} + [\boldsymbol{C}^{(0)}]^{-1} \boldsymbol{R} \tag{39}$$

with constant matrix coefficients $\boldsymbol{K}^{(0)}, \boldsymbol{V}^{(0)}, \boldsymbol{C}^{(0)}$. The eigenvalues of $\boldsymbol{M}_k$ control the behavior of the system in the vicinity of the stationary state. The matrix $\boldsymbol{M}_k$ is non-Hermitian due to the "advection" matrix $\boldsymbol{V}^{(0)}$ and the "phase change" matrix $\boldsymbol{R}$. Thus these two processes can principally cause eigenvalues with positive real parts (causing a growth of perturbations) and



non-zero imaginary parts (causing oscillatory behavior). In the case of $\boldsymbol{V}^{(0)} = \boldsymbol{R} = 0$, Eq. (39) predicts negative eigenvalues
for all $k > 0$ and no pattern formation as it should be for simple diffusion equations.

The matrix $\boldsymbol{M}_k$ can be diagonalized in closed form using the symbolic algebra software MAPLE and eigenvalues subsequently separated into real and imaginary parts. Evaluating the eigenvalues with the corresponding parameter values from Table (1) shows that one of the eigenvalues has positive real part and non-zero imaginary part indicating a wave instability. This is controlled by the matrix $\boldsymbol{V}^{(0)}$ with the coefficients $k_1$ and $D_1$ (cf. Eq. (C14) in appendix (C)) which characterize the
sensitivity of the effective coefficients on density. Thus the instability originates from the fact that the effective diffusivity and/or the effective conductivity of snow depend on ice volume fraction (or snow density) and it is triggered by transitions in snow density, i.e. layers.

We can compare the prediction of the stability analysis with the simulation of the full model for the Gaussian crust by analyzing the growth of amplitude of Fourier modes for wave number $k$ through the Power spectral density of the perturbation.
To this end we have computed the third component of the perturbation vector $u_3^{(1)} = \phi_i(z,t) - \phi_i(z,0)$ as the deviation from the initial, oscillation-free state. The space-time plot of the perturbation in real space is shown in Fig. (6)(a) which shows the emergence of the traveling wave. Discrete time steps are shown in Fig. (6)(b) which displays the self-amplifying behavior of the density modulation at in the layer-transition region i.e. at the boundary of the crust. The Power spectrum $|\tilde{u}_3^{(1)}|^2$ of the perturbation is computed via Fast Fourier transform and shown in (6)(c) as a function of wave number for different times
together with the theoretically predicted range of instability (eigenvalues with positive real part and non-zero imaginary part, grey region) derived from the diagonalization of the matrix $\boldsymbol{M}_k$. The comparison confirms that growing modes are consistently lying in the unstable range, confirming our theoretical explanation how wave-like patterns emerge in density gradient regions as inherent features of the coupled heat and mass transport in snow with its temperature dependent material properties.

## 7 Sensitivity studies

Finally we conducted a few sensitivity studies to facilitate a discussion about the relevance of our findings in future snow modeling.

To confirm that observed wave patterns in the Gaussian crust are robust against variations of absolute values in temperature gradients and initial crust density we simulated a denser Gaussian crust with different values for the temperature gradient. Figure (7)(a) shows that the non-linear PDE system essentially obeys time vs temperature gradient scaling. Virtually the same
wave solution is obtained at different physical times $t$ which scale as $t \sim (\nabla T)^{-1}$. This test also shows that for a higher crust density, the density depletion on the sublimating side is more pronounced.

Second we subjected a smoothly varying density profile to a temperature gradient of 50K/m and conducted a long simulation over 170 days. The results are shown in Fig. (7)(b). This confirms that if the density profile is sufficiently smooth, even an entire season simulation shows a stable, smooth advection of the ice phase.

Finally, we mimic the situation of a snowpack over a dry soil where no vapor can enter the system from below by imposing a zero-flux Neumann BC on the vapor equation. The results are shown in Fig Fig. (7)(c) indicating the order of magnitude of



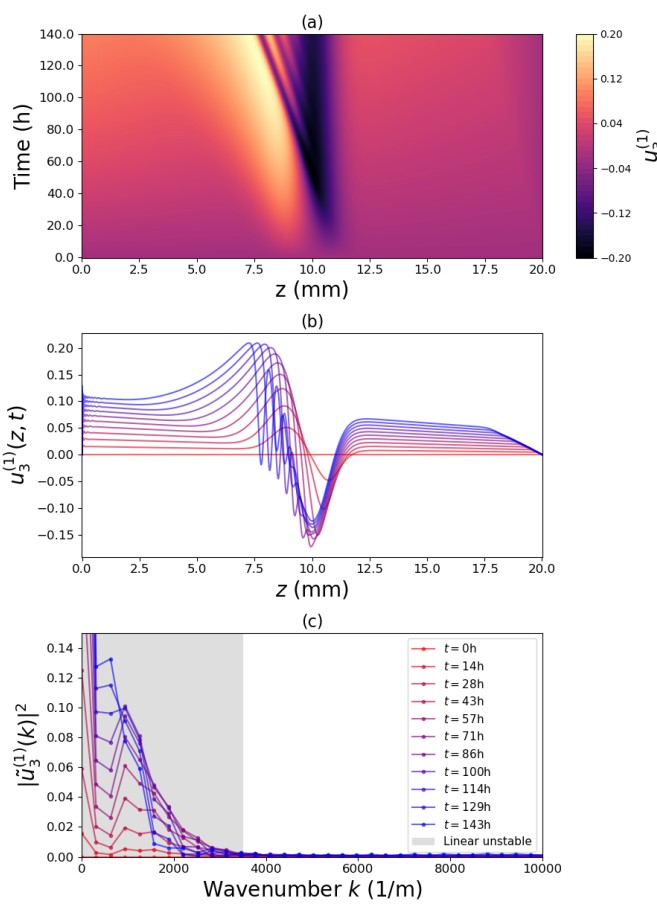

**Figure 6.** Space time plot of the perturbation field(a) profiles at different times steps (b) and corresponding power spectra (c).

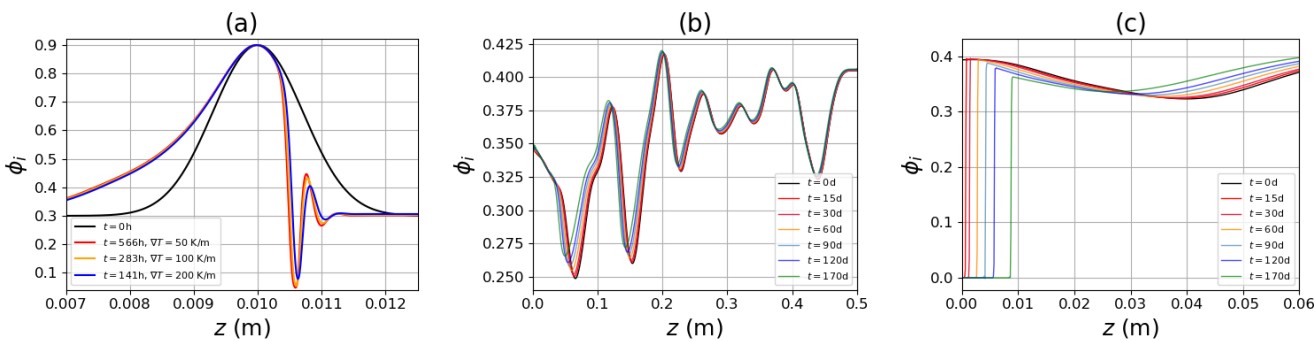

**Figure 7.** Simulation of the fully coupled system: a) time-temperature scaling b) Evolution of a smooth initial ice profile and fixed mean temperature gradient of $50 K/m$. c) Same as b) with homogeneous Neumann BC and impact on depletion of mass at the boundary.




the expected depletion of the snow mass at the bottom for purely diffusive vapor transport under the influence of a temperature gradient of $50K/m$ for an entire season.

## 8 Discussion

### 8.1 Comparison of published homogenization schemes

We have revisited published models of coupled heat and (diffusive) vapor transport in snow. To investigate their numerical requirements we have implemented a stand-alone solver into the open source software FEniCS for the sake of flexibility in numerical experiments involving (spatial and temporal) resolution, solution strategies and accuracy.

From a mere physical point of view, our comparison of (Calonne et al., 2014) and (Hansen and Foslien, 2015) has shown that both schemes yield similar results as long as the same parametrizations are used as closure for the PDE coefficients. The impact of purely diffusive vapor transport on macroscopic density changes is rather small (Fig. (7), in agreement with (Jafari et al., 2020). Accurate estimates will certainly rely on the choice for the parametrization of the transport coefficients which naturally cause differences in the results (Fig. (1)). In homogeneous parts of the snowpack these differences are small, larger differences can be expected at layer transition regions (Fig. (2)) where phase changes sensitively react to the underlying homogenized process model equations.

For the present work, however, the precise numbers were of minor importance. The primary goal was an assessment of the non-linearity that is contained in published vapor homogenization schemes and their numerical requirements in view of previously reported numerical issues Jafari et al. (2020); Adams and Brown (1990); Hansen and Foslien (2015). While we have certainly challenged the numerical scheme by predominantly exploring high temperature and high density gradients, the observed time - gradient scaling (Fig. (7)) indicates that the underlying non-linear mechanisms are generally robust against these details.

### 8.2 Oscillatory behavior in the numerical solution

We have discovered that both models (Calonne et al., 2014) and (Hansen and Foslien, 2015) exhibit oscillations in the numerical solution if they are dynamically coupled to an evolving ice phase (Eq. (7)). From our analysis of different combinations of initial and boundary conditions (Sec. 4) we conclude that two type of oscillations can occur, node-to-node oscillations (indicating numerical problems) and smooth oscillations (features of the underlying equations)

In view of numerical problems, we have shown that the dynamic coupling of the ice phase to heat and mass transport is equivalent to an advection of the ice phase (Sec 5). The non-linear nature of this advection causes a self-amplified increase of density changes/layer transitions, imposing special requirements on mesh resolution, time stepping and potentially shock-capturing schemes (Shu and Osher, 1988) to avoid oscillations. The fact that node-to-node oscillations occur on the "outflow" boundary of the ice phase supports this. In most of our simulations we have imposed a Dirichlet boundary condition of the vapor phase which is strictly in equilibrium on the boundary. By virtue of Eq. (23) this implies that the snow density cannot





change directly at the boundary, while in the interior of the domain the ice is piling up towards the boundary under the advection (Fig. 4) This is reminiscent for outflow boundaries in fluid dynamics for high local Péclet numbers (Donea and

Huerta, 2003): The disagreement between the information transported through the domain from the cold boundary and the information prescribed on the warm boundary is numerically resolved by steep node-to-node oscillations (Fig. 3. Similar issues can arise whenever abrupt changes in the advection velocity cannot sufficiently be resolved on a given mesh, as in Figure 5 in the middle of the domain. This interpretation of node-to-node oscillations is supported by the fact that oscillations can be completely suppressed on the lower boundary by choosing a boundary conditions $\nabla \rho_v = \partial/\partial T \rho_v^{eq} \nabla T$ for the vapor equation,

which implies a vanishing gradient for the ice profile at the boundary. Our tests indicated that node-to-node oscillations can be largely reduced by choosing a high resolution in space and time. A more efficient approach might be the utilization of stabilization methods where our approximation (Eq. (23)) may serve to provide a local advection velocity of the ice phase which needs to be supplied e.g. to a SUPG stabilization scheme (Donea and Huerta, 2003).

As a nasty coincidence, after increasing mesh and time resolution to suppress numerical problems, the true solution con-

verges to a smooth, travelling wave pattern (Fig. 5, Fig. 6). As confirmed by the theoretical analysis, these oscillations are true features of the non-linear homogenization equations and triggered by gradients in the density (Sec.6.2). These patterns can only form when the ice equation is dynamically coupled in a mass conserving way to the heat and vapor equation. At the boundary, these physical oscillations are triggered as a consequence of the Dirichlet boundary condition as explained above: The competition of a fixed value of $\phi_i$ directly on the boundary for $z = 0$ (implication of the imposed boundary conditions on

the vapor phase for Eq. 7) and the increase of $\phi_i$ in the interior of the domain gives rise to a transition layer at the boundary with a high density gradient in the ice phase that triggers the instability according to the mechanism revealed in Sec. 6.2. This is again supported by the fact that the physical oscillations at the boundary can be suppressed by choosing the BC for the vapor equations such that the gradient in ice volume fraction vanishes. This behavior is reminiscent of the wave instability triggered by a Dirichlet boundary condition in a 1D non-linear reaction-advection-diffusion system (Vidal-Henriquez et al., 2017) which

travels *up-stream* into the domain.

### 8.3 Weak-layer formation by wave instability?

We have shown that spatial heterogeneities in the density/ice volume fraction can amplify under the coupled thermodynamic description in snow. This has been pointed out before Adams and Brown (1990). We have investigated this phenomenon within the idealized scenario of a Gaussian crust where density gradients self-amplify under the non-linear advective dynamics with a

subsequent instability and the emergence of wave patterns. The eigenvalues of the linearized PDE system in our linear stability analysis has revealed the mechanism: A non-homogeneous density paired with the density dependence of the effective diffusion coefficients triggers the instability. Pattern formation following an instability is ubiquituous in non-linear (diffusion-reaction) PDEs (Cross and Hohenberg, 1993).

In the present case, the observed instability may have far reaching consequences which comes as a (surprising) side-product

of our numerical study: As the instability causes the depletion of density on the sublimating side of a crust (Fig. 7(a) it explains why a low density (mechanical weak) layer can form under these conditions. For high density crusts, the parameters and model





components used here, predict a considerable reduction of the density in a sub-millimeter layer above the crust (Fig. 7(a)). This mechanism is solely a consequence of the continuum description of snow. We stress, that this is different from a previously suggested reasoning (Colbeck and Jamieson, 2001) and it may occur as a superimposed effect on additional microstructural
controls through near crust metamorphism (Hammonds et al., 2015). In the future, it will be therefore important to assess how this instability is affected by adding mechanical settling, an evolving microstructure or enhanced mass transport from convection.

### 8.4 Open questions

### 8.5 Limits of the homogenization scheme

The present work poses questions on the limits of validity of the used homogenization schemes that are relevant for a user of the equations.

First, as detailed in (Calonne et al., 2014), the homogenized equations (1), (2) are not valid for arbitrary growth rates. This is a tricky situation since the PDE system exhibits self-propelled dynamics into a state of fast growth for high density gradients, thereby leaving its own range of validity. However, since the the homogenization scheme (Calonne et al., 2014) does not
contain the ice phase, it remains unclear if the used equations remain valid if a dynamical ice phase were included in the first place.

Second, the two-scale homogenization (Calonne et al., 2014) states assumptions on length scales, on which the description is meant to be valid. The expansion dictates what can be considered as homogenous, macroscopic scale which is sufficiently large against the microstructural length scale. But what happens if the derived equations contain mathematical features on
smaller scales which violate the separation of scales? We have shown that the wave instability produces patterns on millimeter to sub-millimeter scales requiring ridiculously small mesh sizes to resolve them, clearly interfering with characteristic length scales of the microstructure. What is a consistent way of dealing with this situation? Resolving them numerically and averaging them out? Suppressing them numerically by artificial diffusion? Or does this behavior signal true multi-scale effects where the assumed "seperation of scales" fundamentally breaks down? Answering these questions appears to be a key demand for future
work on homogenization to provide a robust recipe how to use derived equations correctly for applied modeling.

#### 8.5.1 Discontinuous vs continuous description of a stratified snowpack

Any snowpack contains variations in its properties that may be described either by continuous profiles containing large gradients (as done here) or by a discontinuous stacking of layers. We want to point out here that if a continuous description of density variations were to be replaced by discontinuous layers, the investigated wave problem won't disappear. In a hypothet-
ical discontinuous layer setting, as commonly pursued in snowpack models, the mass continuity of the ice phase at the layer interface would require to derive a dynamic equation (likely non-linear) equation for the migration of the layer interface on which the continuity of temperature and heat and mass fluxes were to be imposed. From the continuous description used here, no firm conclusion can be drawn on the behavior of the interface evolution between discontinuous, homogeneous layers. We



hypothesize though, that the wave instability of the continuous PDE formulation may translate into an oscillatory instability
for the position of the interface. In view of the mathematical overhead of tracking continuity conditions at moving interfaces,
we tend to recommend a continuous description in future snow modeling with numerical schemes that cope with arbitrary
gradients in the properties.

### 8.6 Advantages and disadvantages of the numerical framework

We have used FEniCS for the stand-alone implementation to minimize the FEM implementation effort, yet retaining full control
on the numerical solution. Overall FEniCS provides a convenient, modular setup for exchanging PDE coefficients ($k_{\text{eff}}, D_{\text{eff}}$)
or boundary conditions, e.g. for more sophisticated exchange of vapor with the soil or the atmosphere. Alongside our study,
we evaluated the FEniCS framework by comparing numerically different implementations. These experiences are shared for
future reference.

We found that integration by parts in the weak formulation is not only necessary to apply Neumann boundary conditions,
but also increases the precision, regardless of the order of interpolation polynomials. Operator splitting turned out to be of
limited value, the decrease in precision was not outweighed by a decrease in numerical complexity. As expected, a non-
dimensionalization of the equations and the corresponding re-scaling to values of order unity did not impact the solution, as long
as solver convergence settings are adapted. Increasing the polynomial order of the test functions was equivalent to increasing
the mesh resolution with the corresponding number of nodes. However, we experienced large errors, if the polynomial order of
the variables of the coupled equations was not the same, e.g. solving $\phi_i$ in first order, $\rho_v, T$ in second order caused large errors
on the entire domain and also violations of Dirichlet boundary conditions. Adding auxiliary algebraic variables (as done here
for the source terms) can improve the solvability of the system, when convergence settings of the coulped system are adapted.
Using so called sub-functions in the formulation of the variational problem for coding modularity can introduce errors. We
encountered deviations between a sub-function value and its hard-coded counterpart.

While FEniCS has provided an excellent numerical framework for the present study, a clear drawback of FEniCS would
however emerge if mechanical settling was to be considered, as a necessary follow-up extension. In the presence of settling,
the ice phase conservation equation is an advection dominated problem which is notoriously difficult to solve on a Eulerian FE
mesh without numerical smoothing. Remeshing is currently not supported in FEniCS. To this end we have explored another,
fully different numerical route to enable a flexible coupling of transport and phase changes to mechanical settling which is
presented in part 2 of this companion paper (Simson et al., 2021).

## 9 Conclusions

We have shown that the widely accepted form of homogenized vapor transport equations in snow predicts *mathematical* fea-
tures (density waves) with interesting *physical* implications (weak layers) which constitute a considerable *numerical* challenge
for future snowpack modeling.





Combining numerical experiments with theoretical considerations we have shown that the wave instability originates from the dependence of the effective heat and vapor diffusion coefficients ($k_{\text{eff}}, D_{\text{eff}}$) on snow density. Since this dependence is the most fundamental non-linearity in coupled heat and vapor transport in snow, it is unlikely that this effect will luckily disappear when considering more complex, non-linear extensions of the model. The instability is a true feature of published equations and comes at play when the ice phase is dynamically coupled to the vapor phase by phase changes in a mass conserving way.

The instability is triggered by high density gradient regions which either (pre-)exist in snowpacks in the form of layers or which are generated from the self-amplification of density gradients under the coupled dynamics. This amplification is a consequence of the effective advection of the ice phase due to phase changes. This is explained within the derivation of an approximate, non-linear and non-local advection equation. Given the observed (approximate) equivalence between time and imposed temperature gradient, the system always undergoes a self-propelled evolution into its own instability. The instability

might be practically irrelevant as long as smooth density profiles are considered. But the instability will certainly become relevant if a snowpack model should be applicable to simulate a sublimating side of a crust as a potential origin of weak layer formation.

We have outlined open questions and limitations of the present study related to the homogenization scheme, the numerical scheme and the concept of discontinuous layers. While the present study required a stand-alone numerical implementation, it

seems to be of key importance that future snow models will be flexible enough for conducting advanced numerical studies. Only then re-implementations of stand-alone numerical experiments will become obsolete and the rich, non-linear behavior of snow could be predicted from a snow model alone.

*Code and data availability.* Will be made available upon acceptance

## Appendix A: Parametrizations of the PDE coefficients

For the equilibrium vapor pressure we used the parametrization from (Mason, 1971) given by

$$\rho_v^{\text{eq}} = (a_0 + a_1(T - T_m) + a_2(T - T_m)^2)\exp(-T_{ref}/T) \tag{A1}$$

with $T_m = 273.15\text{K}$ and $a_0 = 3.6636 \cdot 10^12, a_1 = -1.3086 \cdot 10^8, a_2 = -3.3793 \cdot 10^6$ and $T_{ref} = 6150\text{K}$.

For the effective heat capacity we used a volume averaged formulation given in (Calonne et al., 2014)

$$(\rho C)_{\text{eff}} = \phi_i \rho_i C_i + \phi_a \rho_a C_a \tag{A2}$$

with $C_i, C_a, \rho_i$ given in Table 1

In the Calonne case we used for the effective diffusion parameters Eq. 12 from Calonne et al. (2011) and the self-consistent estimate used in Calonne et al. (2014) given by

$$k_{\text{eff}}(\phi_i) = k_0 + k_1 \rho_i \phi_i + k_2(\rho_i \phi_i)^2 \tag{A3}$$

$$D_{\text{eff}}(\phi_i) = D_0(3\phi_i - 1)/2 \tag{A4}$$





with $k_0 = 0.024, k_1 = -1.23\,10^{-4}, k_2 = 2.5\,10^{-6}$ and $D_0$ given in Table 1

In the Hansen case we used Eq. (87) and (88) from (Hansen and Foslien, 2015) given by

$$k_{\text{eff}}(\phi_i) = \phi_i\left((1-\phi_i)k_a + \phi_i k_i\right) + \phi_a\left(\frac{k_i k_a}{\phi_i(k_a + LD_0\frac{\partial \rho^{eq}(T)}{\partial T} + (1-\phi_i)k_i)}\right) \tag{A5}$$

$$D_{\text{eff}}(\phi_i) = \phi_i(1-\phi_i)D_0 + \phi_a\left(\frac{k_i D_0}{\phi_i(k_a + LD_0\frac{\partial \rho^{eq}(T)}{\partial T} + (1-\phi_i)k_i)}\right) \tag{A6}$$

$$\tag{A7}$$

and $k_a, k_i, L, D_0$ given in Table 1

## Appendix B: Analysis of residuals

Due to the lack of an analytical solution for the non-linear problem, we used the nodal residuals as an indicator for the solution error. FEniCS does not provide access to the nodal residuals to verify convergence in the `NonlinearVariationalSolver` class. To this end we recovered the residuals manually by decoupling the equations and iterating over individually fine-tuned solvers for each equation. The results are shown in Figure B1. While in this solution scheme the residuals are several orders of

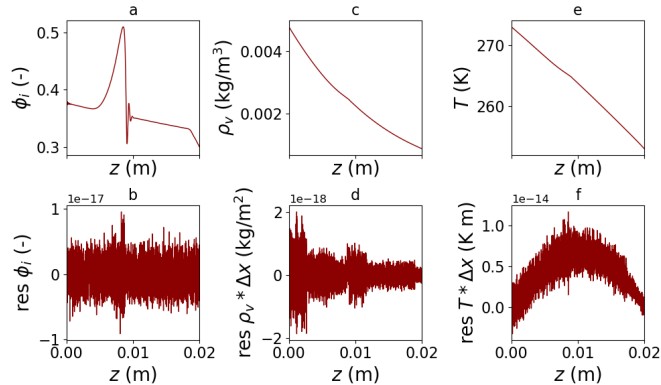

**Figure B1.** Results upon lowering the residuals of all equations. While some pattern is still visible, the order of magnitude is small enough to assume small errors on the solution as well.


magnitude smaller than in our regular setup, the onset of the wave instability remains the same (Figure B2).

## Appendix C: Perturbation expansion of the heat-vapor-ice system

Carrying out the perturbation expansion of the PDE system (30) around the stationary state (31) using the ansatz (37) requires to derive two auxiliary relations which are stated below.


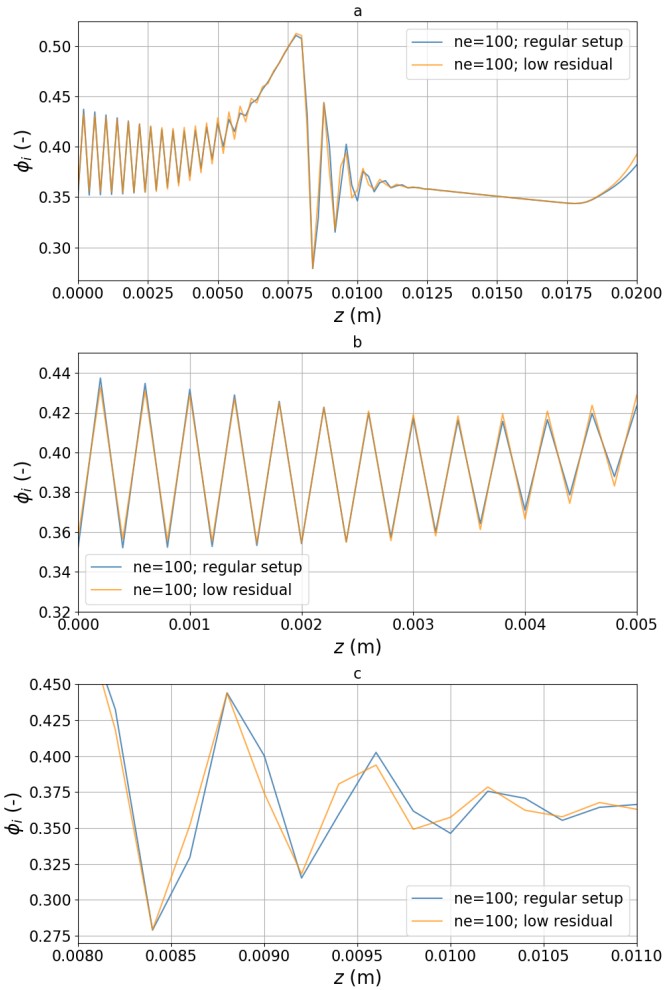

**Figure B2.** Comparison between node-to-node oscillation patterns for the regular setup and the low-residual setup.

First, for any matrix $M = M(u)$ that depends only on the third component $u$ (ice volume fraction $\phi_i$) the following holds

$$\nabla_z M(u) = (e_3 \cdot \nabla_z u) \frac{\partial M(u)}{\partial \phi_i} \tag{C1}$$

where $(\cdot)$ denotes the scalar product on the 3D $(T, \rho, \phi)$ solution space.

Second, the expansion around the fixed point of an arbitrary matrix $M = M(u)$ that depends only on the third component $u$ (ice volume fraction $\phi_i$) reads

$$M(u) = M(u^{(0)}) + (e_3 \cdot u^{(1)}) \frac{\partial M(u)}{\partial \phi_i} \Big|_{u=u^{(0)}} \tag{C2}$$

$$= M^{(0)} + (e_3 \cdot u^{(1)}) M^{(1)} \tag{C3}$$

$$\tag{C4}$$





where we use the following shorthand notation for the matrix coefficients

$$\boldsymbol{M}^{(n)} = \left.\frac{\partial^n \boldsymbol{M}(\boldsymbol{u})}{\partial \phi_i^n}\right|_{\boldsymbol{u}=\boldsymbol{u}^{(0)}} \tag{C5}$$

Now we can state the expansion of the PDE system (30) to first order in $u$ as follows:

$$
\left[\boldsymbol{C}^{(0)} + (\boldsymbol{e}_3 \cdot \boldsymbol{u}^{(1)})\boldsymbol{C}^{(1)}\right]\left[\partial_t \boldsymbol{u}^{(0)} + \partial_t \boldsymbol{u}^{(1)}\right]
$$
$$
- \left[\boldsymbol{K}^{(0)} + (\boldsymbol{e}_3 \cdot \boldsymbol{u}^{(1)})\boldsymbol{K}^{(1)}\right]\left[\nabla_z^2 \boldsymbol{u}^{(0)} + \nabla_z^2 \boldsymbol{u}^{(1)}\right]
$$
$$
- \left[(\boldsymbol{e}_3 \cdot \nabla_z \boldsymbol{u}^{(0)}) + (\boldsymbol{e}_3 \cdot \nabla_z \boldsymbol{u}^{(1)})\right]\left[\boldsymbol{K}^{(1)} + (\boldsymbol{e}_3 \cdot \boldsymbol{u}^{(1)})\boldsymbol{K}^{(2)}\right]\left[\nabla_z \boldsymbol{u}^{(0)} + \nabla_z \boldsymbol{u}^{(1)}\right] = \boldsymbol{R}\boldsymbol{u}^{(0)} + \boldsymbol{R}\boldsymbol{u}^{(1)} + \boldsymbol{f} \tag{C6}
$$

Equating zero and first order terms in $\boldsymbol{u}^{(0)}$ provides an equation that is satisfied by the stationary state as it should be. By collecting terms linear in $\boldsymbol{u}^{(1)}$ the governing equation reads

$$\boldsymbol{C}^{(0)}\partial_t \boldsymbol{u}^{(1)} - \boldsymbol{K}^{(0)}\nabla_z^2 \boldsymbol{u}^{(1)} - \left[(\boldsymbol{e}_3 \cdot \nabla_z \boldsymbol{u}^{(1)})\right]\left[\boldsymbol{K}^{(1)}\right]\left[\nabla_z \boldsymbol{u}^{(0)}\right] = \boldsymbol{R}\boldsymbol{u}^{(1)} \tag{C7}$$

The advection term can be re-written in the form $\boldsymbol{K}^{(1)}\left[\nabla_z \boldsymbol{u}^{(0)} \otimes \boldsymbol{e}_3\right]\nabla_z \boldsymbol{u}^{(1)}$ and after multiplying the corresponding matrices we arrive at the final, linear PDE system with constant coefficients

$$\boldsymbol{C}^{(0)}\partial_t \boldsymbol{u}^{(1)} - \boldsymbol{K}^{(0)}\nabla_z^2 \boldsymbol{u}^{(1)} - \boldsymbol{V}^{(0)}\nabla_z \boldsymbol{u}^{(1)} = \boldsymbol{R}\boldsymbol{u}^{(1)} \tag{C8}$$

with coefficient matrices given by

$$\boldsymbol{C}^{(0)} = \begin{bmatrix} (\rho C)_{\text{eff}}(\phi_{i,0}) & 0 & 0 \\ 0 & (1-\phi_{i,0}) & 0 \\ 0 & 0 & 1 \end{bmatrix} \tag{C9}$$

$$\boldsymbol{K}^{(0)} = \begin{bmatrix} k_{\text{eff}}(\phi_{i,0}) & 0 & 0 \\ 0 & D_{\text{eff}}(\phi_{i,0}) & 0 \\ 0 & 0 & 0 \end{bmatrix} \tag{C10}$$

$$\boldsymbol{V}^{(0)} = \begin{bmatrix} 0 & 0 & k_1\frac{T_H-T_0}{H} \\ 0 & 0 & D_1\rho_1^{\text{eq}}\frac{T_H-T_0}{H} \\ 0 & 0 & 0 \end{bmatrix} \tag{C11}$$

$$\boldsymbol{R} = \begin{bmatrix} -L\alpha\rho_1^{\text{eq}} & L\alpha & 0 \\ \rho_i\alpha\rho_1^{\text{eq}} & -\rho_i\alpha & 0 \\ -\alpha\rho_1^{\text{eq}} & \alpha & 0 \end{bmatrix} \tag{C12}$$

$$\boldsymbol{f} = \begin{bmatrix} L\alpha\rho_1^{\text{eq}}T_{\text{ref}} - L\alpha\rho_0^{\text{eq}} \\ -\rho_i\alpha\rho_1^{\text{eq}}T_{\text{ref}} + \rho_i\alpha\rho_0^{\text{eq}} \\ \alpha\rho_1^{\text{eq}}T_{\text{ref}} - \alpha\rho_0^{\text{eq}} \end{bmatrix} \tag{C13}$$





In the latter step we employed the expansions of the diffusion coefficients around the reference volume fraction $\phi_{i,0}$ in the form

$$k_{\text{eff}} = k_0 + k_1(\phi_i - \phi_{i,0})$$
$$D_{\text{eff}} = D_0 + D_1(\phi_i - \phi_{i,0})$$

$$\text{(C14)}$$

Generally, it seems feasible (though tedious) to carry out an expansion of (C8) in terms of the eigenfunctions of the differential operator satisfying the BC to incorporate the BC in the stability analysis (since $\boldsymbol{u}^{(0)}$ already satisfies the original Dirichlet conditions, the perturbation $\boldsymbol{u}^{(1)}$ must vanish on the boundaries). However, we limit ourselves here to the simpler case of a stability analysis in an infinite domain and take continuous Fourier transforms of (30) with respect to $z$. Denoting the Fourier
variable by $k$ we end up with the final result stated in Eq.(38)

*Author contributions.* The study was designed by HL and JK. Implementation, simulations, and figures were done by KS with contributions and supervision from HL and JK. Derivation of the quasi-advection and the linear stability analysis was done by HL. The manuscript was written by HL with contributions from KS and JK.

*Competing interests.* The authors declare that they have no conflict of interest

*Acknowledgements.* We thank Andy Hansen for discussions and sharing experiences on his numerical implementation.



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
