# Peer review of "Elements of future snowpack modeling - part 1: A physical instability arising from the non-linear coupling of transport and phase changes"

_The Cryosphere, 2021_

## Author Comment (AC1)

Dear Reviewer,

thank you very much for the encouraging feedback and the suggestions for improvements. Below please find your comments in black with our inline replies in red.

Sincerely, on behalf of the authors

Henning Löwe

Schurholt et al. show that coupled equations for heat transport, vapour diffusion and ice mass conservation in snow permit wave solutions in density. The linear stability analysis is nice work that, together with numerical solutions of the nonlinear equations, demonstrates that these are true mathematical solutions and not numerical artefacts. The setting is limited to be somewhat short of a full snow thermodynamics model, and the question of how mm-scale waves in solutions of the continuous equations relate to a bicontinuous material with mm-scale structure remains open.

Specific comments by line number:

5

Is FEniCS widely enough known to name in an abstract without explanation?

In fact, no. Changed.

16

No physically based snow model would neglect vapour transport between snow and the atmosphere in its mass balance. What is commonly neglected is internal vapour transport in the snow (which does not directly influence overall mass balance) and vapour exchange with the soil.

We agree that this may be misleading. Specification "internal" added.

107

What value is used for Beta? Calonne et al (2014) describes its measurement as a challenge.

The value for beta has been added in the table and this difficulty has been pointed out again.

Table 1

Units of vapour pressure are incorrect, and this should be vapour density. Incorrect units for D0. Use scientific notation in place of 2e-5.

This is supposed to be *density*. Changed.

172 (and hereafter)

Set vector u in boldface italic.

Changed.

185

Superscripts n and n+1 should be inside the parentheses on the lhs of equation 9.

Corrected.

219

could note H = 1 m

Noted.

220

The description in Calonne et al. (2014) is much easier to follow than equation (13): the surface temperature decreases linearly from 273K at t=0 to 263K after 5 hours and then remains constant.

Why is T at z = 1 m only slightly below 270K after 10 hours in Figure 1?

There was an error in the figure legend. It is not the solution after 10h that is compared here. After 10h the solution is a stationary (linear) temperature profile for all cases which is meaningless to compare. Corrected.

Figure 1 caption

Transient temperature decrease at the boundary, not an increase

Corrected.

Condensation rate would be a more intuitive profile to show in place of "rhs energy eq.".

Changed.

229

Hansen and Fosllien (2015) envisaged this as a snowpack containing an ice crust. The solid ice at the base of the snowpack was imposed to prevent vapour entering from below.

Description adapted.

244

No comparison is made with tomography experiments, so why choose such a small snow depth?

The goal of this scenario is to explore the behavior of the PDE system when the density changes on small length scales. As outlined in the discussion, each layer transition in a snowpack constitutes such a situation. Another example might be thin crusts as studied in the given reference by X-ray tomography. And since the solution far away from the crust is well behaved (and "boring"), it is sufficient to work with a small depth. Description adapted.

250

Incorrect units of sigma^2.

Corrected.

252

300K snow in Figure 3 is passed without comment. A full snow model (and, indeed, nature) would not permit this.

Agreed and comment added.

255

Advection of the ice crust by sublimation and deposition was already apparent in Scenario 2.

Formulation changed.

283

Is there a missing ice density in equation 24?

Yes. Corrected.

300

Deff *is* linear in ice volume fraction for the Calonne model.

Yes but k_eff is not constant so this is still different from the Calonne model.

305

The oscillations at the boundary in Figures 3 and 4 are clearly numerical artefacts and are not the ones of interest in the following. They are reminiscent of instabilities in an unstable numerical solution of the linear advection equation and could be controlled (as actually shown in 6.1).

Agreed. But we are not classifying the type of oscillations here, we are just saying that we care about all oscillations in further detail in the next section.

310

What were ne and dt in Figure 4? What is the time in Figure 5? Why are the oscillations on the sublimating side of the crust not apparent in Figures 3 and 4?

Number of elements and time step, notation adapted to match the text. The oscillations are not visible in Fig 3 because it shows the solution at an earlier physical time. For Figure 4 we believe that the instability is actually removed when making the approximations that lead to the pure advection equation (23). However, the behavior of the latter is difficult to analyze analytically.

Figure 5

Units of dt should be given in the legends.

Corrected.

444

Why is this a "nasty coincidence"?

This is rephrased now. But it remains nasty, since both type of oscillations (numerical and physical) appear almost hand in hand while adding complexity to the innocent question "how does vapor transport change the mass balance in a snowpack".

550

Vapour density is required

Corrected.

553

Error in exponent for a0 value. All of these parameters have units.

Corrected.

Minor corrections:

25

"have been used for a long time"

Corrected.

31

"revisited the problem"

Corrected.

49

Richards equation

Corrected.

61

"design"

Corrected.

179

"implementation in"

Corrected.

346

"PDE system (26)"

Corrected.

383

"density modulation in the layer-transition region"

Corrected.

407

"a stand-alone solver in the open source software"

Corrected.

534

"comes into play"
Corrected.

**Citation**: https://doi.org/10.5194/tc-2021-72-RC1

---

## Author Comment (AC2)

Dear Reviewer,

thank you very much for the encouraging feedback and the suggestions for improvements. We appreciate the explicit suggestions for language improvements. Below please find your comments in black with our inline replies in red.

Sincerely, on behalf of the authors

Henning Löwe

Review on "Elements of future snowpack modeling – part 1: A physical instability arising from the non-linear coupling of transport and phase changes" by Konstntin Schürholt, Julia Kowalski, and Henning Löwe.

The paper describes consequences of the incorporation of vapor transport in snowpack models on the numerical schemes in idealized settings. The water vapor transport in combinations with phase changes introduces additional non-linear terms in the heat and mass balance equations which the standard numerical schemes do not have to take care for. In three case studies the non-linear differential equation system is solved with a python-based Finite Element Framework for 1D snowpack models.

It is shown convincingly that currently used continuum-mechanical models derived through homogenization or mixture theory yield similar results for homogeneous snowpacks of constant density. However, if the snow density and temperature varies significantly with depth phase changes result in non-linear advection of the ice phase. This advection potentially exaggerates density variations which potentially initiates wave instability in the continuity equations. A linear stability analysis reveals that the wave instabilities are caused by the density dependence of the effective transport coefficients.

The work deserves publication. The presented analyses are sound and the reasoning behind the several steps of the analyses is clear.

However, the quality of the presentation of the results is poor. This holds for the textual form but also for the readability of the figures. The sentence construction is often unnecessary complicated making reading of the manuscript a tedious work. Below I will present some examples and suggest corrections but the list is not complete. I recommend to get help from a native English speaker.

line 4: Spell out PDE

Done.

line 5: Skip 'solely'

Done.

line 7: Skip 'different,'. Take care of coma.

Done.

line 8: "For heterogeneous situations in which the snow density varies significantly with depth, we show that phase changes in the presence of temperature gradients give rise to a non-linear advection of the ice phase that amplifies existing density variations." I suggest: "When snow density varies significantly with depth, we show that phase changes in the presence of temperature gradients give rise to non-linear advection of the ice phase amplifying existing density variations."

Changed accordingly.

line 18: "As hypothesized in recent work on shallow tundra snowpacks (Barrere et al., 2017; Domine et al., 2016) persistent temperature gradients throughout the season may contribute to the depletion of snow density at the bottom of the snowpack due to persistent upward vapor fluxes." I suggest:

"Persistent temperature gradients throughout the season may contribute to the depletion of snow density at the bottom of the snowpack due to upward vapor fluxes, as has been hypothesized for shallow tundra snowpacks by Barrere et al. (2017) and Domine et al. (2016)."

Changed accordingly.

line 31: "Lastly (Hansen and Foslien, 2015) was revisiting the problem of coupled heat and vapor transport using mixture theory which led to a more restrictive set of transport equations that rely on the assumption that the vapor concentration is always close,but not exactly in equilibrium with temperature. While the existing vapor schemes largely differ in the form of the effective transport coefficients, there is a general agreement on the basic type and form of the partial differential equations (PDE), that govern coupled heat and diffusive vapor transport in snow." I suggest: "Hansen and Foslien (2015) revisited the problem of coupled heat and vapor transport using mixture theory leading to a more restrictive set of transport equations. They rely on the assumption that the vapor concentration is always close, but not exactly in equilibrium with temperature. While the existing vapor schemes largely differ in the form of the effective transport coefficients, there is a general agreement on the basic type and form of the partial differential equations (PDE) governing coupled heat and diffusive vapor transport in snow."

Changed accordingly.

line 42: "The first attempt to solve the vapor diffusion equation in a snowpack model was recently undertaken by (Jafari et al.,2020) who equipped the model SNOWPACK with a vapor transport scheme as a non-linear reaction-diffusion equation." I suggest: "Recently Jafari et al. (2020) equipped the model SNOWPACK with a vapor transport scheme in form of a non-linear reaction-diffusion equation. It is the first attempt to solve the vapor diffusion equation in a snowpack model."

Changed accordingly.

line 43: "The numerical solution requires time-steps of 1 min and mesh sizes of 1 mm to avoid "numerical oscillations" that were observed, even within an implicit, unconditionally stable numerical scheme." I suggest: "Even within an implicit, unconditionally stable numerical scheme, "numerical oscillation" requires very small time-steps of 1 min and mesh sizes of 1 mm."

This would change the meaning. We therefore stick to the previous version.

line 50: "Phase change processes in seasonal and polar snowpacks are commonly of interest on long time scales ideally using coarse meshes and large time steps to meet requirements for climate modeling." I suggest: "Phase change processes on seasonal time scales in polar snowpacks are important for climate modeling ideally adequately simulated on coarse meshes with long time steps"

Changed accordingly.

line 51: "It is therefore necessary ..." -> "Therefore, it is necessary ..."

Changed accordingly.

line 62: "It is the aim of the present paper to advance ..." -> "We aim to advance ..."

Not changed.

line 73: "This is confirmed by an analytical, linear stability analysis which relates unstable behavior to the density dependence of the effective (heat and mass) diffusion constants. The results suggest that previously observed oscillations in the numerical treatment (Adams and Brown, 1990; Jafari et al., 2020) were not numerical problems but may rather have hypoallergenic physics." I suggest: "This is confirmed by an analytical linear stability analysis attributing the unstable behavior to the density dependence of the effective heat and coefficients. The results suggest that previously obtained oscillations in the numerical schemes (Adams and Brown, 1990; Jafari et al., 2020) are physical and not numerical artifacts."

Changed accordingly. But "hypoallergenic physics" (auto-correction?) is reasonably funny....

line 97: "$\rho\_i$ is the density of ice density (assumed to be constant)" -> "$\rho\_i$ is the constant density of ice"

Changed.

line 110: "in (Calonne et al., 2014)" -> "in Calonne et al. (2014)"

Corrected.

line 117 to 123: I suggest to rewrite the paragraph. I could not understand it.

The paragraph has been improved.

line 125: "the same is not true" -> rewrite

Rewritten.

line 128: "into a single one that no longer" -> rewrite

Rewritten.

line 136: "was not been considered" -> "is not considered"

Corrected.

line 140 to 143: I wonder why here the models are referenced as 'Calonne' and 'Hansen' but not in the paragraph in line 124 to 129. Homogenize it.

Not clear what is meant here. We actually do refer to the model as "Hansen" in line 126...

line 161: "In summary, all symbols and parameter values used in this study are provided in Table 1." -> "We summarize all symbols and parameter values in Table 1."

Changed.

line 171: "For the spatial discretization we note that the non-linear PDE systems, of interest can be formally rewritten in the form" -> "The non-linear PDE systems of interest can be rewritten in the form"

Corrected.

line 177: You should explain what 'small support' means.

Explanation has been added.

line 190: "The vapor equation has by far the fastest dynamics, followed by the energy.190The ice mass balance instead has a much slower dynamics." I suggest: "The vapor equation has by far the fastest dynamics, followed by the second fasted, the energy equation. The ice mass balance equations has a much slower dynamics."

Changed.

line 191 to 193: Hardly understandable sentence. Rewrite.

Sentenced has been improved.

line 200: "... which is known to be stable and converge of second order for linear operators." Do you mean "and convergent at second order"?

Yes. Sentence corrected.

line 209: Add 'we': "For each of the three physical scenarios we evaluate three model formulations:"

Changed.

line 214: "The first scenario is taken from (Calonne et al., 2014) who investigated the response of a homogeneous snow layer to transient heating. To this end we use the IC" I suggest: "The first scenario is proposed by Calonne et al. (2014) and investigates the response of a homogeneous snow layer on transient heating. The initial conditions are:"

Changed accordingly.

line 224: "For this combination of IC and BC we obtain the results in Figure (1) where the solutions of all three cases at t= 10h are shown." I suggest: "The solutions of all three cases we obtain for this combination of IC and BC are shown in Figure 1 for t=10h."

Changed accordingly.

Figure 1 and Figure2: The lines can be hardly identified. I suggest to increase the line width.

Agreed.

Figure 3: Same here. If the lines are to close to each other I suggest to write that in the figure caption.

Agreed again.

line 302: "The fact that Deff decreases, and keff increases with ice volume fraction, decreases the ice flux functional G in high density regions over lower density regions." This sentence is not understandable.

The sentence has been rephrased.

line 316: "This is interesting, as it suggests that these waves are true, intrinsic features of the full Calonne model equations, rather than an artifact of the numerical scheme." I suggest: "This suggests that these waves are an intrinsic features of the Calonne model equations ruthes than an artifact of the numerical scheme."

Changed accordingly.

line 320: "To comprehend the oscillatory nature of the solution we analyzed the problem theoretically within perturbation theory." I suggest: "We use perturbation theory to comprehend the oscillatory nature of the solution."

Changed accordingly.

Figure 5: Same as for the other figure. Increase in width and if lines overlay mention it in the caption.

Done.

line 335: Skip "To the end"

Skipped.

line 336: "small" -> "thin"

Corrected.

line 337: "can be always" -> "is always"

Changed.

line 245 Is (28) correct?

It's supposed to be Eq (26). Corrected

line 386 to 388: Split the sentence in several sentence. Hardly understandable.

We agree. Improved.

line 444: "As a nasty coincidence". Very sloppy language.

Agreed and deleted.

line 479: "the the"

Corrected.

line 496: "equation ... equation"?
Corrected.

**Citation**: https://doi.org/10.5194/tc-2021-72-RC2